# Evaluation of Reference Genes for Real-Time Quantitative PCR Analysis in Tissues from Bumble Bees (*Bombus Terrestris*) of Different Lines

**DOI:** 10.3390/ijms232214371

**Published:** 2022-11-19

**Authors:** Kathannan Sankar, Hyung Joo Yoon, Young Bo Lee, Kyeong Yong Lee

**Affiliations:** Division of Apiculture, Department of Agricultural Biology, National Institute of Agricultural Science, Rural Development of Administration, Wanju-gun 55365, Republic of Korea

**Keywords:** bumble bee, reference gene, qRT-PCR, gene expression, RefFinder

## Abstract

Bumble bees are important alternative pollinators and model insects due to their highly developed sociality and colony management. In order to better understand their molecular mechanisms, studies focusing on the genetic and molecular aspects of their development and behavior are needed. Although quantitative real-time polymerase chain reaction (qRT-PCR) can be used to quantify the relative expression of target genes, internal reference genes (which are stably expressed across different lines and tissues) must first be identified to ensure the accurate normalization of target genes. In order to contribute to molecular studies on bumble bees, we used *Bombus terrestris* to determine the expression stability of eight reference genes (β-actin (ACT), Arginine Kinase (AK), Phospholipase A2 (PLA2), Elongation factor 1 alpha (EF-1), Ribosomal proteins (S5, S18, S28) and glyceraldehyde-3-phosphate dehydrogenase (GAPDH)) in five different lines and several tissues (ovary, thorax, fat body, and head) using RT-qPCR procedures and four analysis programs (RefFinder, NormFinder, BestKeeper, and geNorm). In general, the S28, S5, and S18 ribosomal protein genes and the PLA2 and EF-1 genes showed the highest stability and were therefore identified as suitable reference genes for the bumble bee species and their defined lines and tissues. Our results also emphasized the need to evaluate the stability of candidate reference genes for any differently designed lines and tissue conditions in bumble bee species.

## 1. Introduction

The bumble bee (*Bombus terrestris*) plays an important role as a pollinator [1]. In addition, the bumble bee is a prominent key model insect due to its relatively complex behaviors, such as sociality, division, and colony management. This makes the bumble bee a prominent key model insect [2]. The earlier studies have demonstrated that endocrine system status and gene expression are play significant factors for flexible bumble bee colony management, which involves colonies of different lines regulating their queen division and population dynamics. By examining the expression patterns of the genes allegedly linked to reproduction management across various bumble bee lines and tissues, it is possible to increase our understanding of the molecular mechanisms that underlie the regulation of bumble bee colony physiology [3].

Real-time PCR, also known as quantitative PCR (qPCR), has become one of the most widely used standard methods for the sensitive and highly sequence-specific detection and quantification of mRNA expression. qPCR is a powerful and ubiquitous method for tracking relative changes in gene expression [4,5]. In order to guarantee reliable results from this technique, a normalization procedure is required [6,7,8]. Assessing the target gene to an endogenous control (reference gene) in the same sample is a classic and common strategy. At the present, housekeeping genes are universally used as a reference [9]. Therefore, selecting stable reference genes is absolutely necessary in order to determine the transcript changes of target genes using quantitative real-time PCR [10]. In addition to this, it is essential to determine which reference genes are appropriate for each unique situation. There are a few different algorithms, such as geNorm [11,12]. BestKeeper [13] and NormFinder [14,15], are utilized in the process of candidate reference gene evaluation. The frequently used housekeeping genes, such as those encoding β-actin (ACT), phospholipase (PLA2), arginine kinase (AK), glyceraldehyde-3-phosphate dehydrogenase (GAPDH), S18 or S28 ribosomal RNA, and elongation factors (EF-1) are commonly used as suitable internal control genes. Although housekeeping genes are generally involved in the maintenance of cellular homeostasis, they are assumed to be constitutively expressed. However, many studies have shown that the levels of expression of these genes vary widely according to a variety of factors [16,17,18,19]. Housekeeping genes, such as actin or ribosomal genes, are thought to be universal for critical cell functions and to be produced constitutively and steadily under a variety of physiological and experimental circumstances. Although, recent research has shown that the expression levels of housekeeping genes may differ based on the gene, tissue type, and experimental approach. For instance, it has been discovered that the stability of GAPDH (glyceraldehyde-3-phosphate dehydrogenase), one of the most often used housekeeping genes, varies depending on the kind of tissue [20,21] or metabolic process [22]. Therefore, a comprehensive study of the gene expression of any target gene requires confirming the stability of the normalizing gene of choice in different lines and tissues.

To the best of our knowledge, the RT-qPCR analysis of common housekeeping genes’ stability over time in different lines and tissues of reproductive organs has never been performed. In this study, we obtained stable reference genes of different lines of bumble bee queens (line 1–line 5) and four tissue (ovary, fat body, thorax, and head) types, resulting in eight candidate reference genes. The stability of the reference genes was assessed using five different statistical approaches described in previous literature. Our work identified EF-1, S18, PLA2, GAPDH, and S5 as the most stably expressed genes, while AK, S28, and ACT were found to fluctuate among the different lines, tissues, and reproductive organs. Furthermore, the transcript levels of one target gene, Vg, were used to validate the effectiveness of the selected reference genes.

## 2. Materials and Methods

### 2.1. Sample Preparation and RNA Extraction

The five different lines of *B. terrestris* virgin queens used in this study were kept in the National Institute of Agricultural Science RDA’s insect pollinator laboratory in Wanjangu, Jeonju, Korea (35°44′43″ N, 128°02′22″ E). From the artificial breeding system [23,24], queen virgin bees were collected from five different lines in spring (2022), line 1, line 2, line 3, line 4, and line 5, and each queen bee was selected within 24 h of emergence. Each line was selected in the direction of a high colony formation rate and worker bee productivity and was selected as a line that had maintained the genetic traits of 10 generations or more. The collected queens were immediately flash-frozen in liquid nitrogen and kept at a temperature of 70 °C until RNA extraction for tissue analysis could be performed. After separating the head, thorax, and abdomen of each bee, these body parts were placed in a tube containing TRI reagent and prepared for three separate replications. After that, RNA samples were prepared from the three biological replicates that had been created. After completely homogenizing each tissue sample with a bullet blender, total RNA was extracted using a DNase RNA kit (LOT. 10162017, GenDEPOT, N Fry Rd #301, Katy, TX USA). During the process of RNA extraction, the samples were treated with DNase I so that any traces of genomic DNA could be removed. This was carried out in accordance with the protocol provided by the manufacturer [14]. The purity and quantity of the extracted RNAs were measured in triplicate using a Sartorius Stedim spectrophotometer (Sartorius Stedim Biotech GMBH, Park Blenheim Road, Epsom Surrey, UK). The prepared RNA was then stored at −70 °C until further use.

### 2.2. Candidate Genes and Primer Design

Information regarding the eight commonly used housekeeping genes that were selected for this study can be found in Table 1. Additionally, information regarding the target genes (Vg) that were used for further validation and the selected primer pairs that were used for amplification can be found in Table 1. With the assistance of the Primer-BLAST software, each set of primers was developed in accordance with the MIQE guidelines (NCBI). The coding region of a gene was broken up into portions ranging from approximately 200–400 base pairs (bp), and NCBI software was used to design primers for each of those individual portions. This allowed for the selection of the most effective primers. Following the removal of the primer pairs that, according to NCBI Blast, had the potential to generate amplicons that were not intended for the target, the remaining primer pairs were put through qPCR testing, and the best pair for each gene was chosen for the experiment.

### 2.3. RNA Isolation and cDNA Synthesis

Total RNA was extracted from each biological sample using the RNeasy kit (LOT. 10162017, GenDEPOT, N Fry Rd #301, Katy, TX, USA) according to the manufacturer’s instructions (on-column genomic DNA digestion was performed per the instructions), and RNA concentration and absorbance ratios (A260/280 and A260/230) were measured using a Viva Spec+ spectrophotometer (SMB80-3004-81; S. No. 116000, Sartorius Stedim Biotech, Park Blenheim Road, Epsom Surrey, UK). A total of 250 ng of RNA from each sample was reverse-transcribed in a total volume of 20 µL using amfiRivert cDNA Synthesis Platinum Master Mix (LOT. R5600-100, GenDEPOT, N Fry Rd #301, Katy, TX USA) to produce DNA, which was then analyzed spectrophotometrically and diluted to 100 ng/µL. Next, individual master mixes with each of the DNA primer combinations (e.g., different lines (1–5) and tissues (ovary, body, thorax, and head)), were created for three technical replicas, and the mixtures were distributed onto the qPCR plates (8 μL per reaction well).

### 2.4. Quantitative Real-Time PCR

In this case, qPCR was performed using an Applied Bio system, QuantStudio5, connected apparatus (Thermo Fisher Scientific, Singapore). The reactions were carried out in triplicate using intercalating dye SYBR Green-based PCR Master Mix (LOT. 2203E, BiONEER, Daejeon, Republic of Korea), following the manufacturer’s instructions. Each reaction was performed in a final volume of 9 μL, and primers were used at a concentration of 250 nM. qRT-PCRs were performed in triplicate (technical replicates) using the following protocol: 40 cycles of 95 °C for 15 s, 58 °C for 15 s, and 72 °C for 30 s.

The cycle threshold (Ct) values of the eight candidate reference genes and the target gene (Vg) were obtained at the same fluorescence threshold (0.1). Validation of the reference genes was carried out in order to verify the reliability of the results based on the most stably and least stably expressed reference genes from the stability assessment results. The gene stabilities and the expression level of the target gene (Vg) were determined. The Ct values for reference genes and Vg were obtained for each sample normalized by a relative quantification method adapted from the original 2^−ΔΔCt^ concept. Reference genes were selected based on the stability value ranking of the genes analyzed by geNorm. When multiple references were used for normalization, the mean Ct value was used for analysis.

### 2.5. Evaluation of Reference Gene Expression Stability

RefFinder was used to assess the stability of the potential reference genes [25]. An online analytical tool called RefFinder combines four methods: geNorm [11], NormFinder [14], BestKeeper (version 1) [13,26], and Delta-Ct [27]. Each gene was given a weight based on the rankings produced by each method, and the geometric means of the gene weights were calculated for a thorough final ranking. Candidate genes with a lower mean weight are assumed to be transcriptionally stable and serve as excellent reference genes.

### 2.6. Statistical Analyses

The Ct values of the genes in each of the lines and tissues were analyzed by using OriginPro 2022, and the M, SD, and CV values (CV = AM/SD) for BestKeeper were obtained [25]. In the analysis, Ct values were entered both directly and geometrically. The program then calculated the arithmetic mean, the standard deviation, and the coefficient of variance based on the order in which the genes were ranked in terms of their degree of stability. For NormFinder [14]. In the analysis, the Ct values were first converted to a linear scale, and then the normalization factor was determined by calculating the geometric mean of the candidate reference genes that were a part of the dataset. The geNorm software investigation was carried out by computing the expression stability measure in accordance with the parameters defined in the geNorm paper [11]. Following the analysis of pairwise variation, the genes were ranked according to the positions they occupied. After that, highly correlated candidate genes were combined, and the resulting *p* values were analyzed. According to the findings of BestKeeper, the candidate genes with relatively high R^2^ values but low SDs, CVs, and *p* values were regarded as being the genes with the highest degree of genetic consistency. The M value for each putative reference gene is automatically calculated by geNorm based on the geometric mean of all genes that have been studied; genes with lower M values are more stable than genes with higher M values. After that, a statistical analysis of the relative transcript levels of Vg was performed using SPSS for Windows (version 23.0). This analysis involved calculating the values for each sample using the 2^−ΔΔCT^ method [26], and a one-way analysis of variance (ANOVA) was followed by Tukey’s multiple comparison post hoc test in the statistical examination of the eight candidate reference genes.

## 3. Results

### 3.1. Amplification Specificity and Efficiency

While performing the qRT-PCR analysis, the efficiency and specificity of the amplification process were investigated. Each of the qRT-PCR products that were amplified with each primer set exhibited a single band in 1% agarose gels, whereas RT-PCR detected a distinct single peak in the melting curve for each sample. (Appendix A). In addition, specific primer pairs were designed for the candidate reference genes (ACT, AK, PLA2, EF-1, S5, S18, S28, and GAPDH), and the length of amplicon sequences ranged from 70 to 200 bp (Table 1). Analysis of the melting curve revealed a single peak for every reaction (Appendix A). qPCR was used to investigate expression in different bombus lines and tissues. In our investigation of PCR efficiency, each of the eight candidate genes had linear regression coefficients R^2^ > 0.99, and overall amplification efficiencies ranged from 91–104%.

### 3.2. Cycle Threshold (Ct) Values and Expression Analysis of the Eight Reference Genes

Across all of the different lines and tissues, the Ct values generated in the RT-qPCR ranged between 14 and 35. A low Ct value indicates high expression [28,29] (Figure 1). The S18 gene exhibited the lowest Ct values, ranging from 17 to 22, for all integrated samples, and AK showed the highest Ct values in all integrated samples, ranging from 26 to 35, and the gene with the second highest ranking was S5, i.e., from 30 to 35 (Figure 1). GAPDH exhibited the second lowest Ct values of 14–25, and the Ct values of the genes ACT, EF-1, and PLA2 ranged from 17–33 (Figure 1). In order to perform the reference gene validation study, we verified and compared the expression levels of each reference gene among different lines, tissues, and reproductive organs (Figure 1).

### 3.3. Analysis of Expression Stability Using Four Programs

#### 3.3.1. NormFinder Analysis

Stability values [14] were calculated by NormFinder to determine the best reference gene based on the expression variation of the candidate genes. AK was the most stable gene (mean value = 0.213) accordance with the typical values of stability (mean values) calculated arithmetically for the five different lines (Figure 2A). In a comparison of all eight genes’ stability values, GAPDH was the most stable gene in line 1. EF-1 was the most stable gene in line 2, whereas PLA2 was the most stable gene in line 3; ACT was the most stable gene in line 4, and S5 was the most stable gene in line 5 (Figure 2A, Table 2). In gene stability analysis of the different tissue and reproductive organ types, EF-1 was the most stable (mean stability = 0.319) (Figure 3B). In the stability analysis of the different tissue types, *S18* was the highest ranked gene in the ovary, compared with EF-1 and ACT in the thorax and head (least stable), and the most stable expression levels in the ovary and thorax were S18 and S5, respectively (Figure 2, Table 2). However, according to the results of NormFinder, when the stability values of genes were determined by merging the five distinctive lines and four tissue types, the order of stability ranking from the most (lowest value) to the least (highest value) stable was as follows: S5 > S18 > EF-1 > S28 > PLA2 > GAPDH > AK > ACT (Figure 2C, Table 2).

#### 3.3.2. BestKeeper Analysis

The BestKeeper algorithm selects the candidate reference genes with the lowest values of SD (usually < 1), which are the results of the calculation of the stability of candidate reference genes performed by BestKeeper [13]. BestKeeper identified *S18* (in lines 1 and 2), *S28* (in line 3), *ACT* and *GAPDH* (in lines 4 and 5) as the most appropriate reference genes with the least Ct variation. These conclusions were reached on the basis of the SD and CV values. (Table 3, Figure 3). AK was the top-ranked gene in the body and ovary, according to the SD and CV values, whereas *ACT* was the gene with the lowest stability in the thorax and head, and S18 was the gene that was identified as the optimal reference gene in the thorax and head. This was the case regardless of the different tissue and reproductive organ types. (Table 3). In spite of the fact that the stability of genes in the head, thorax, and fat body of the bumble bees varied, the SD for all eight genes was <1.0 across all tissues. This means that any of the genes could be used as a reference gene to normalize how the target gene is expressed in the bumble bee’s ovary, head, or thorax (Table 3). Whenever the Ct values of the eight genes were applied in combination all over lines and tissue types, Best-Keeper indicated that *S18*, *EF-1*, and *S28* had SD values of less than 1.0. As a consequence, these genes have the potential to be the best candidates for reference genes (Table 3, see the integrated sample).

#### 3.3.3. GeNorm Analysis

The values of the average expression stability (M values) of the eight candidates were also calculated by geNorm across the various lines, tissues, and reproductive organs (Figure 4). M values of 1.5, indicating that it was the stable expression criterion. In the line comparison, the M values of *PLA2* were 1.5 in each of the lines, whereas the other seven genes had M values of 1.5 in at least one season (Figure 4A). Consequently, *PLA2* may be the most suitable reference gene for target gene normalization based on the analysis of the gene expression trends in the different bumble bee lines. When the M scores of the candidate reference genes were compared across different tissues and reproduction organs, all genes had M 1.5, with the least stable being *AK* and *ACT* in the thorax and fat body, head, and ovary (Figure 4B). Therefore, *ACT*, PLA2, EF-1, GAPDH, S28, and S18 have the potential to serve as useful reference genes for the analysis of gene expression in various types of bumble bee tissue. These genes are listed in alphabetical order. When the Ct values of the eight genes that were obtained from the various lines and tissue types were combined, the M values of all of the genes were equal to 1.5. This indicates that the M values of all of the genes are functionally equivalent. According to the findings of the geNorm study, this demonstrates that any one of the eight genes can perform the function of a reference gene.

#### 3.3.4. RefFinder Analysis

Finally, RefFinder was used as a comprehensive analysis tool. In recent decades, it has been commonly applied for determining gene expression stability when geNorm, NormFinder, and BestKeeper results are ambiguous. The overall score of the specific reference genes and the geometric mean of their weights were calculated based on the rankings from geNorm, NormFinder, and Ct. Finally, we evaluated the expression stability of all candidate reference genes in various tissue lines and reproductive organs through using RefFinder algorithm. *S5* and *S18* were found to be the most stably expressed genes across all integrated samples, as well as the different lines and tissue types. The *AK* gene had the most stable expression level in the tissues in the head, compared with *S18* in the ovary and *S5* in the thorax, whereas *EF-1* was the most stably expressed gene in the fat body and among the different lines. In contrast to *AK*, *ACT* was the least stably expressed gene in the integrated sample. Regarding the overall tissue sample, *ACT* and *AK* were the least stably expressed genes (Figure 5, Table 4) based on the results of geNorm, NormFinder, and BestKeeper.

### 3.4. Validation of Reference Gene

We compared the expression levels of Vg (as the target gene) normalized with each of the eight reference genes and multiple reference genes across the various types of lines, tissues, and reproductive organs because the geNorm analysis values suggested the application of multiple reference genes for target gene normalization (Figure 6 and Figure 7). The number of reference genes that were chosen in each of the different line analyses did not have an effect on the amount of Vg that was expressed (Figure 6A–E). In addition, the levels of Vg expression that were normalized with a single gene (for example, ACT, EF1, S5, S18, or GAPDH) did not significantly differ from the levels of Vg expression that were normalized with multiple reference genes (*p* = 0.994 for lines 1 and 2; *p* = 0.998 for lines 3 and 4; *p* = 0.997 for line 4; and 0.952 for line 5). (Figure 6A–E).

These findings suggested that a single gene, such as ACT, EF-1, S5, S18, or GAPDH, could be used as a reference for investigating the expression profiles of target genes in a number of different bombus queen lines. In a comparison of Vg expression in tissues, while ACT, EF-1, S5, S18, or GAPDH were selected as specific reference genes, the Vg expression levels were not substantially different from those acquired with a combination of multiple reference genes in the head (*p* = 0.867), thorax (*p* = 0.941), and ovary (*p* = 0.645). The analysis of the fat body revealed opposite findings (*p* = 0.0163), shown in Figure 7A–D. In addition to this, we contrasted the total expression levels of Vg after normalizing them with a single candidate reference gene and with a combination of multiple genes. (Figure 6F). According to the findings of the investigation, there was not a significant difference between the expression levels of Vg when normalized with any number of multiple gene combinations. Therefore, it would be conceivable to employ a single gene as the reference gene for target gene normalization across many lines and tissue types. Some examples of genes that could be used for this purpose include ACT, EF-1, S5, S18, or GAPDH. This assumes that the gene expression analysis relied on selecting optimal reference genes.

## 4. Discussion

In order to identify optimal reference genes for qRT-PCR assays in different bumble bee lines and tissues, we conducted a genome-wide search. We evaluated the expression stabilities of eight candidate reference genes, ACT, EF-1, PLA2, S5, S18, AK, S28, and GAPDH, using four analysis programs: NormFinder, BestKeeper, geNorm, and RefFinder. The normalization methods, including individual reference genes, were also validated by analyzing the Vg expression in the different lines and tissue samples. The mean Ct values of all of the tested reference genes were between 14 and 35 across the different lines and tissues; a low Ct value indicates high expression [29,30]. Among the candidate genes, S18 had the highest transcript level, indicated by the lowest mean Ct value of 17 to 22, whereas AK had the highest Ct values in the integrated sample, ranging from 26 to 35.

The primers for the eight candidate reference genes seemed to have single peaks in the dissociation curve, revealing that the primers’ specificity met the requirements. (Appendix A). An accurate, optimized qPCR assay should have a linear correlation coefficient (R^2^) and a high PCR amplification efficiency of 91 to 104% [31,32]. In addition, the reference genes that were employed for analysis presented a comparatively wide range of expression levels, from the lowest mean Ct value of 14.4 for GAPDH to the highest of 26–35 for AK, consistent with previous studies [32,33]. According to the findings of our study, the genome of the *B. terrestris* can be amplified by making use of any of the primers that were utilized in the qRT-PCR analysis. In gene expression investigations, reference genes such as GAPDH and ACT are typically chosen because they are constitutively expressed, participate in fundamental housekeeping tasks that are required for cell maintenance, and are therefore frequently chosen as reference genes for normalization purposes [34]. Recent findings from a number of studies have shown that the expression of these genes can be changed in response to a wide variety of stimuli and in the context of pathological circumstances in a variety of tissues while the tissues are growing and differentiating [35]. Consequently, investigation into alternative genes is essential. Popular algorithms for assessing the stability of reference genes from a list of potential reference genes in different lines and tissues include comprehensive ranking, geNorm, NormFinder, BestKeeper, the comparative Ct method, and NormFinder and geNorm.

In the current study, the different algorithms, NormFinder, BestKeeper, and geNorm, produced different results when ranking gene stability, as observed in previous studies [36,37]. Therefore, using all of the algorithms would result in more believable findings if they were integrated. When assessing gene expression in various lines and tissues of reproductive organs, the majority of the eight potential genes might be appropriate for use as reference genes. This would depend on the specific context of the study. Despite the fact that the vast majority of the research that came before this one did not establish a threshold for the level of gene stability in the NormFinder analysis [38], one recent studies have suggested that 0.5 is an appropriate cutoff [39]. Based on this criterion, all eight genes were suitable reference genes according to NormFinder. This result was supported by the distribution of the Ct values, which indicated that all eight candidate genes were stably expressed with CV values < 1, which is considered to indicate low variance. In BestKeeper analysis, all eight genes were also determined to be appropriate reference genes for the analysis of bumble bee gene expression in different lines and tissue, as indicated by SD values < 1. In contrast, when using M 1.5 as the criterion, which has been widely suggested as an acceptable level for reference gene selection [40], all eight genes could be regarded as reference genes across different lines and tissue types. Our combined analyses suggest that EF-1, S5, S18, and GAPDH would be the most suitable as optimal reference genes for the normalization of target gene expression in bumble bee samples prepared from a variety of tissues across different lines.

In the current investigation, it was discovered that the expression levels of Vg that were normalized by either EF-1 or PLA2 were much greater than those that were normalized using ACT, EF-1, S5, S18, or GAPDH. This was the case across a variety of cell lines and tissue types. In addition, the Ct values of EF-1 and PLA2 were significantly greater compared to those of the other six genes. As a consequence of our findings, the genes EF-1, PLA2, S28, S5, S18, and GAPDH have been proposed for consideration as potential reference genes for qRT-PCR analysis. The standard deviation values of ACT and AK were higher than 1.0, which was the cutoff line used in BestKeeper. This was despite the fact that other analyses showed that the expression stability values of these six genes were lower than the requirements. Therefore, in conclusion, the stability values of S5, S18, GAPDH, and EF-1 were below the cutoff values in each of the analysis methods used. As a result, we propose S5, S18, GAPDH, and EF-1 as the best reference genes for accurate normalization of the expression of genes in bumble bee samples prepared from various tissues and lines. Five statistical algorithms (RefFinder, geNorm, NormFinder, BestKeeper, and the ΔCt method) are usually applied to analyze the stabilities of candidate reference genes. However, inconsistent results obtained from the above programs were found in previous studies and in this research. For the accurate evaluation of reference genes, a comprehensive method named RefFinder was used in the final evaluation, which has been used for many species [41,42]. We also conducted a comprehensive evaluation selection of candidate reference genes based on these five programs (geNorm, NormFinder, BestKeeper, ΔCt method, and RefFinder). As a case in point, we performed qPCR on the five different lines of bumble bees and the tissue markers (Figure 7) and demonstrated that the choice of reference genes should be based on an accurate description of the gene expression of target genes throughout the reprogramming process. This study supported precise results by utilizing the proper reference qualities to track down the overall expression levels in *B. terrestris* lines and tissues. To our knowledge, this is the first report to provide a detailed investigation of the potential candidate reference genes in different lines of bumble bees. We hope that this study will encourage further examination and will encourage the community to perform systematic investigations of appropriate reference genes prior to gene expression analyses.

## Figures and Tables

**Figure 1 ijms-23-14371-f001:**
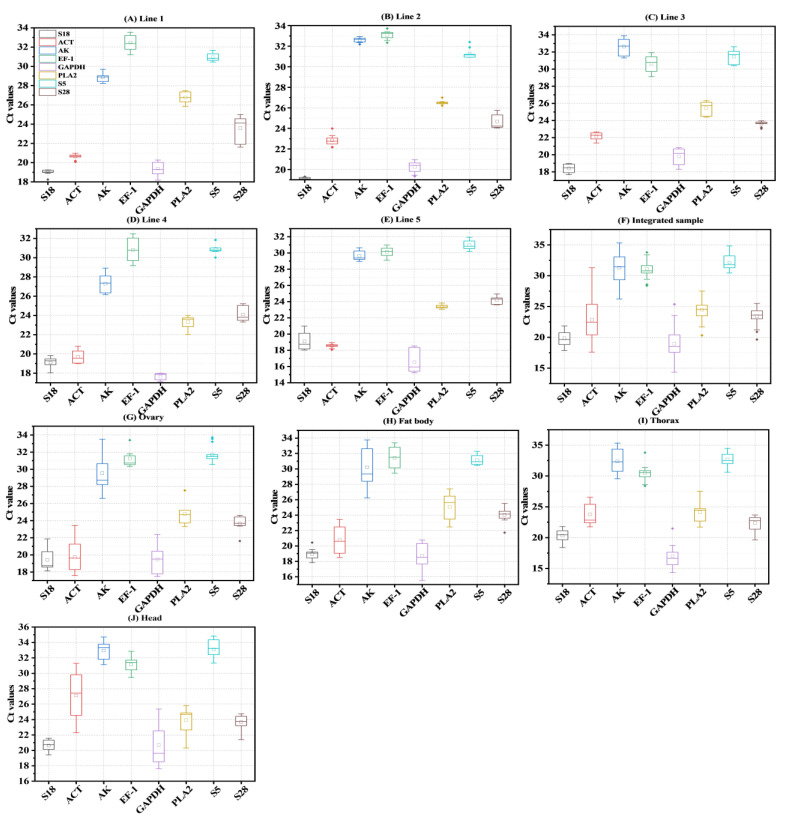
A box-and-whisker plot depicting the Ct value range of the eight candidate reference genes in the bumble bee samples. *B. terrestris* line samples ((**A**) line 1, (**B**), line 2, (**C**), line 3, (**D**), line 4, (**E**), line 5)), (**F**), Integrated samples of *B. terrestris*. ((**G**) head, (**H**) thorax, (**I**) fat body, (**J**) ovary)), Tissues and reproductive organs samples of *B. terrestris.* The values of three biological replicates are presented as the average of three technical replicates. The whiskers represent the n-sample standard deviation. The error bars represent the highest and lowest values. The dots represent outliers (replicated samples with Ct values greater than 50% of the interquartile range).

**Figure 2 ijms-23-14371-f002:**
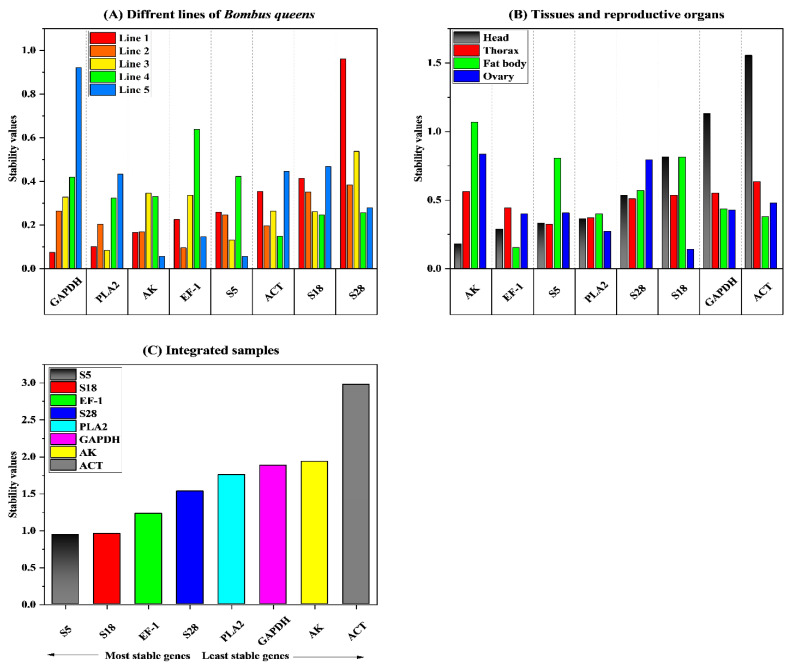
NormFinder was used to calculate the expression stability values of the eight candidate reference genes. The average stability values (mean values) of the bumble bee samples prepared from five different lines (**A**), the different tissue and reproductive organ types (**B**), and an integration of all samples were calculated arithmetically (**C**).

**Figure 3 ijms-23-14371-f003:**
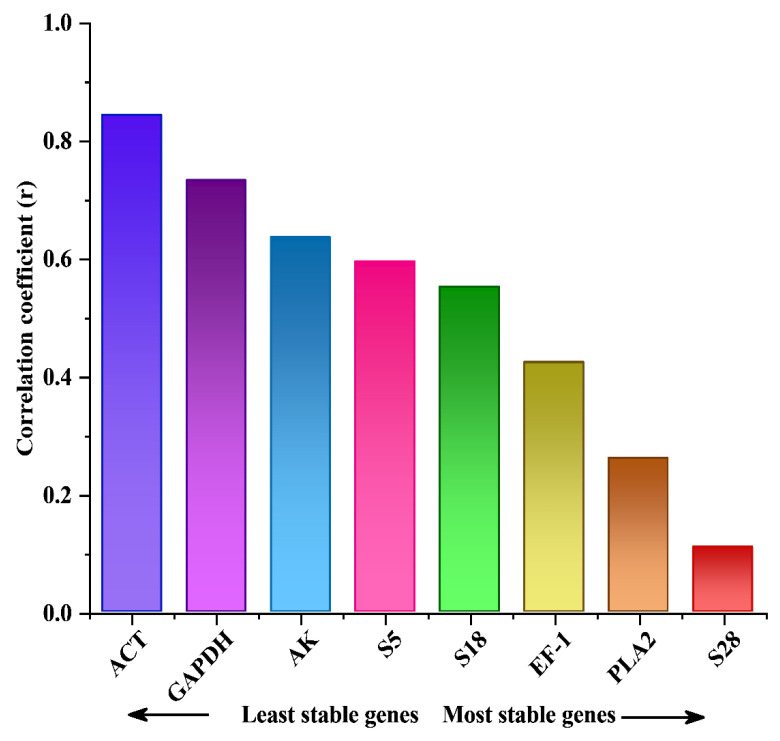
The BestKeeper index and the level of reference gene expression are related. BestKeeper was used to examine the stability of the eight reference genes.

**Figure 4 ijms-23-14371-f004:**
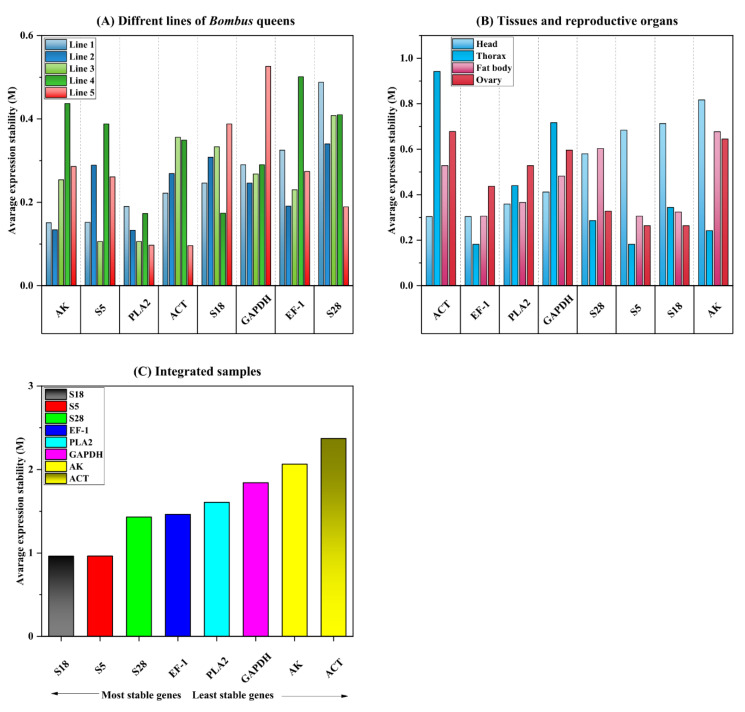
GeNorm calculated the expression stability values (M) of the eight-candidate bumble bee reference genes. Five different lines (**A**), four tissue and reproductive organ types (**B**), and an integration of all samples were used to prepare samples (**C**).

**Figure 5 ijms-23-14371-f005:**
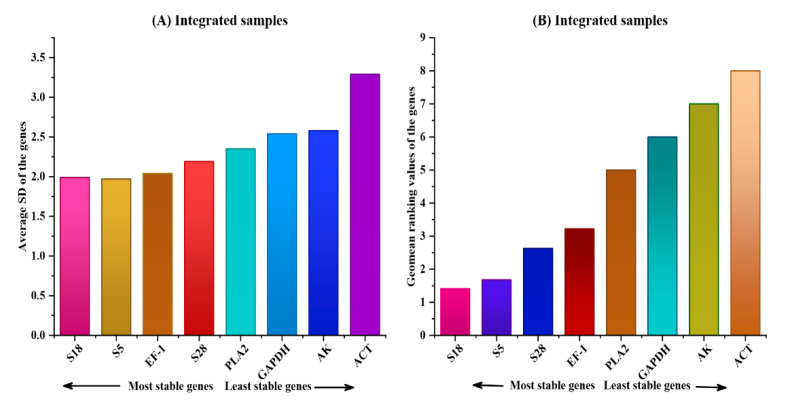
Ranking of eight reference genes by the web-based RefFinder tool. (**A**) Stability ranking of the eight reference genes analyzed by the ΔCt method presented as the average SD. (**B**) Comprehensive stability ranking of the eight reference genes presented as the geomean of the gene values.

**Figure 6 ijms-23-14371-f006:**
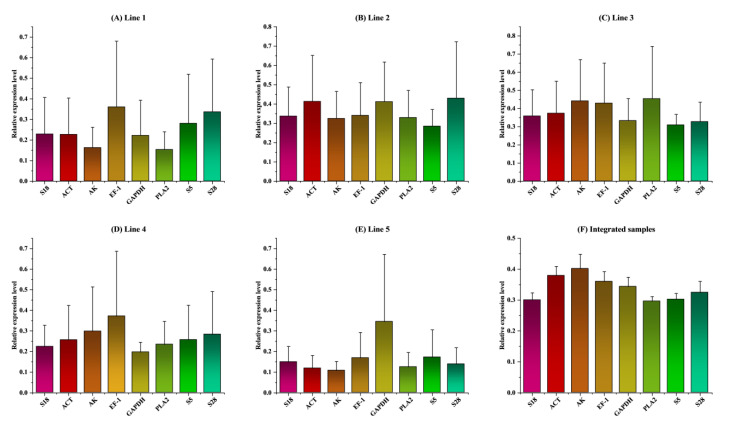
Variations in the levels of expression of the genes that could function as references in different lines. *B. terrestris* line samples ((**A**) line 1, (**B**) line 2, (**C**) line 3, (**D**) line 4, (**E**) line 5)), (**F**), Integrated samples of *B. terrestris*. The expression levels of these genes in the bumble bee samples were standardized with respect to the Ct values of Vg, which is an internal target gene. The graphs present the relative mRNA expression values that were determined by the application of the 2^−ΔΔCt^ technique. Comparing the groups at a significance level of 0.05 required the use of a one-way analysis of variance (ANOVA), followed by a post hoc analysis that included multiple comparisons. The values that are displayed represent the mean ± SE.

**Figure 7 ijms-23-14371-f007:**
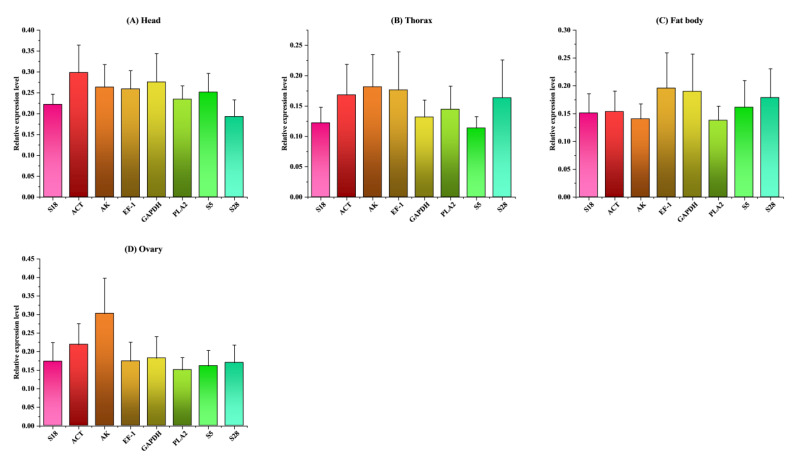
Changes in the levels of expression of the candidate reference genes in a variety of tissues and reproductive organs samples of *B. terrestris.* ((**A**) head, (**B**) thorax, (**C**) fat body, (**D**) ovary)), The levels of gene expression in the bumble bee samples were compared to the Ct values of Vg, an internal target gene, in order to achieve a normalized reading. The values of the relative mRNA expression that were determined through the use of the 2^−ΔΔCt^ method are depicted in the graphs. Comparing the groups at a significance level of 0.05 required the use of a one-way analysis of variance (ANOVA), followed by multiple comparisons using post hoc analysis. The data are presented with the mean ± SE.

**Table 1 ijms-23-14371-t001:** Primer sequences and accession numbers of the eight candidate reference genes and the target gene in bumble bees. F: forward; R: reverse; TM: annealing temperature.

Gene Symbol	Gene Description	Gene Bank ID	Primer Sequences (5′-3′)	Product Length (bp)	TM (°C)
*S18*	s18 Ribosomal	XM_003400778.3	F “AGCGTGCTGGAGAATGTTCA”	101	59
			R “TCGTTCCAAGTCCTCACGAAG”		
*ACT*	Actin	XM_003396942.3	F “CGACTACCTCATGAAGATT”	101	59
			R “CGACAACAAAGTTTCTC”		
*AK*	Arginine Kinase	XM_003401454.4	F “CACACGAGGTTCACTGCTCT”	183	59
			R “GGAGAAGCCAGCTTCCAGTT”		
*EF-1*	Elongation factor 1 alpha	XM_012314816.3	F “GAGAAGTGCGCCGCTAGT”	94	59
			R “AACGCGAATTAAGCGGATGC”		
*GAPDH*	Glyceraldehyde-3-phosphate	XM_003398087.3	F “GCTGGAGCTGAATATGTTGTAGAATC”	195	59
			R “AGTAGTGCAGGAAGCATTAGAGATAACT”		
*PLA2*	Phospholipase A2	XM_003400908.4	F “TTGCGATGCGCATGACATTT”	114	59
			R “ATCGCAGCCGATTGATACCC”		
*S5*	s5 Ribosomal	XM_012308701.2	F “TGCGCAATGTCCTATAGTCG”	157	59
			R “AGCCGTCACAAGAACCTGTA”		
*S28*	s28 Ribosomal	NW_025963548.1	F “AGCGCGGATATCTTGGACTG”	83	59
			R “TCAAGACGGGTCCTGAGAGT”		
Vg	Vitellogenin	XM_012308109.3	F “AAGAATCATCTGAGCAACGTGA”	106	59
			R “TAGTGCACTGTTTGCTTTTGGT”		

**Table 2 ijms-23-14371-t002:** Delta Ct (SD) ^a^ refers to the standard deviation, which is shown in Figure 1, and is a summary of the expression stability values of the eight genes analyzed by Ct distribution in NormFinder, BestKeeper, and geNorm. (SV) ^b^ denotes the stability values determined by NormFinder. The standard deviation of the Ct values was analyzed by BestKeeper and is indicated by (SD) ^c^ (SV) ^d^ denotes the stability values determined geNorm.

**Rank**	**Line 1**				**Line 2**			
	**Delta-Ct (SD) ^a^**	**NormFinder (SV) ^b^**	**Bestkeeper (SD) ^c^**	**geNorm (MV) ^d^**	**Delta-Ct (SD)**	**NormFinder (SV)**	**Bestkeeper (SD)**	**geNorm (MV)**
1	*S18* (0.53)	*S18* (0.13)	*S18* (0.20)	*EF-1|S5* (0.306)	*S18* (0.08)	*EF-1* (0.096)	*S18* (0.08)	*AK|PLA2* (0.133)
2	*PLA2* (0.58)	*PLA2* (0.27)	*ACT* (0.22)	*EF-1|S5* (0.306)	*PLA2* (0.15)	*AK* (0.169)	*PLA2* (0.46)	*AK|PLA2* (0.133)
3	*EF-1* (0.6)	*EF-1* (0.39)	*S5* (0.34)	*S18* (0.324)	*AK* (0.22)	*ACT* (0.197)	*AK* (0.22)	*EF-1* (0.191)
4	*S5* (0.61)	*S5* (0.40)	*AK* (0.35)	*PLA2* (0.366)	*EF-1* (0.35)	*PLA2* (0.204)	*EF-1* (0.35)	*GAPDH* (0.246)
5	*GAPDH* (0.64)	*GAPDH* (0.42)	*PLA2* (0.45)	*GAPDH* (0.482)	*S5* (0.38)	*S5* (0.247)	*S5* (0.47)	*ACT* (0.269)
6	*ACT* (0.67)	*ACT* (0.47)	*GAPDH* (0.61)	*ACT* (0.528)	*ACT* (0.46)	*GAPDH* (0.265)	*ACT* (0.15)	*S5* (0.289)
7	*S28* (0.87)	*S28* (0.79)	*EF-1* (0.70)	*S28* (0.603)	*GAPDH* (0.47)	*S18* (0.352)	*GAPDH* (0.38)	*S18* (0.308)
8	*AK* (0.9)	*AK* (0.83)	*S28* (1.24)	*AK* (0.677)	*S28* (0.63)	*S28* (0.384)	*S28* (0.63)	*S28* (0.34)
**Rank**	**Line 3**				**Line 4**			
	**Delta-Ct (SD)**	**NormFinder (SV)**	**Bestkeeper (SD)**	**geNorm (MV)**	**Delta-Ct (SD)**	**NormFinder (SV)**	**Bestkeeper (SD)**	**geNorm (MV)**
1	*PLA2* (0.32)	*PLA2* (0.085)	*S28* (0.22)	*PLA2|S5* (0.106)	*ACT* (0.41)	*ACT* (0.149)	*GAPDH* (0.33)	*S18|PLA2* (0.173)
2	*S5* (0.33)	*S5* (0.132)	*ACT* (0.41)	*PLA2|S5* (0.106)	*S18* (0.43)	*S18* (0.247)	*S5* (0.42)	*S18|PLA2* (0.173)
3	*ACT* (0.39)	*S18* (0.261)	*S18* (0.42)	*EF-1* (0.23)	*S28* (0.45)	*S28* (0.257)	*S18* (0.49)	*GAPDH* (0.29)
4	*S18* (0.39)	*ACT* (0.265)	*PLA2* (0.71)	*AK* (0.254)	*PLA2* (0.47)	*PLA2* (0.323)	*PLA2* (0.56)	*ACT* (0.349)
5	*GAPDH* (0.42)	*GAPDH* (0.328)	*S5* (0.71)	*GAPDH* (0.268)	*AK* (0.49)	*AK* (0.331)	*ACT* (0.61)	*S5* (0.388)
6	*AK* (0.42)	*EF-1* (0.337)	*AK* (0.81)	*S18* (0.333)	*GAPDH* (0.52)	*GAPDH* (0.42)	*S28* (0.71)	*S28* (0.41)
7	*EF-1* (0.42)	*AK*(0.346)	*EF-1* (0.84)	*ACT* (0.356)	*S5* (0.54)	*S5* (0.423)	*AK* (0.75)	*AK* (0.437)
8	*S28* (0.56)	*S28* (0.538)	*GAPDH* (0.85)	*S28* (0.408)	*EF-1* (0.69)	*EF-1* (0.638)	*EF-1* (1.05)	*EF-1* (0.501)
**Rank**	**Line 5**				**Integrated sample**			
	**Delta-Ct (SD)**	**NormFinder (SV)**	**Bestkeeper (SD)**	**geNorm (MV)**	**Delta-Ct (SD)**	**NormFinder (SV)**	**Bestkeeper (SD)**	**geNorm (MV)**
1	*S5* (0.38)	*S5* (0.056)	*ACT* (0.21)	*ACT|PLA2* (0.097)	*PLA2* (0.83)	*PLA2* (0.207)	*S28* (0.81)	*S18|S5* (0.255)
2	*AK* (0.4)	*AK* (0.056)	*PLA2* (0.21)	*ACT|PLA2* (0.097)	*EF-1* (0.87)	*EF-1* (0.554)	*S18* (0.95)	*S18|S5* (0.255)
3	*EF-1* (0.42)	*EF-1* (0.147)	*S28* (0.38)	*S28* (0.189)	*S5* (0.88)	*GAPDH* (0.56)	*EF-1* (0.98)	*EF-1* (0.282)
4	*S28* (0.44)	*S28* (0.279)	*S5* (0.50)	*S5* (0.261)	*S18* (0.93)	*S5* (0.567)	*S5* (1.04)	*S28* (0.383)
5	*PLA2* (0.5)	*PLA2* (0.433)	*EF-1* (0.52)	*EF-1* (0.274)	*S28* (1.03)	*AK* (0.685)	*PLA2* (1.19)	*PLA2* (0.436)
6	*ACT* (0.51)	*ACT* (0.446)	*AK* (0.56)	*AK* (0.286)	*GAPDH* (1.08)	*S18* (0.713)	*GAPDH* (1.83)	*GAPDH* (0.665)
7	*S18* (0.62)	*S18* (0.468)	*S18* (0.90)	*S18* (0.388)	*AK* (1.14)	*S28* (0.844)	*AK* (1.92)	*AK* (0.783)
8	*GAPDH* (0.94)	*GAPDH* (0.921)	*GAPDH* (1.26)	*GAPDH* (0.526)	*ACT* (2.06)	*ACT* (2.017)	*ACT* (2.76)	*ACT* (1.103)
**Rank**	**Ovary**				**Fat body**			
	**Delta-Ct (SD)**	**NormFinder (SV)**	**Bestkeeper (SD)**	**geNorm (MV)**	**Delta-Ct (SD)**	**NormFinder (SV)**	**Bestkeeper (SD)**	**geNorm (MV)**
1	*S18* (0.53)	*S18* (0.139)	*S28* (0.53)	*EF-1|S5* (0.306)	*EF-1* (0.63)	*EF-1* (0.152)	*S18* (0.47)	*ACT|EF-1* (0.304)
2	*PLA2* (0.58)	*PLA2* (0.27)	*S5* (0.75)	*EF-1|RPS5* (0.306)	*ACT* (0.71)	*ACT* (0.376)	*S5* (0.51)	*ACT|EF-1* (0.304)
3	*EF-1* (0.60)	*EF-1* (0.398)	*EF-1* (0.78)	*S18* (0.324)	*PLA2* (0.72)	*PLA2* (0.396)	*S28* (0.65)	*PLA2* (0.359)
4	*S5* (0.61)	*S5* (0.404)	*PLA2* (0.93)	*PLA2* (0.366)	*GAPDH* (0.74)	*GAPDH* (0.433)	*EF-1* (1.25)	*GAPDH* (0.412)
5	*GAPDH* (0.64)	*GAPDH* (0.425)	*S18* (1.01)	*GAPDH* (0.482)	*S28* (0.79)	*S28* (0.567)	*GAPDH* (1.36)	*S28* (0.58)
6	*ACT* (0.67)	*ACT* (0.476)	*GAPDH* (1.36)	*ACT* (0.528)	*S5* (0.91)	*S5* (0.804)	*ACT* (1.39)	*S5* (0.684)
7	*S28* (0.87)	*S28* (0.793)	*ACT* (1.39)	*S28* (0.603)	*S18* (0.91)	*S18* (0.812)	*PLA2* (1.47)	*S18* (0.713)
8	*AK* (0.9)	*AK* (0.833)	*AK* (1.69)	*AK* (0.677)	*AK* (1.13)	*AK* (1.066)	*AK* (1.96)	*AK* (0.817)
**Rank**	**Thorax**				**Head**			
	**Delta-Ct (SD)**	**NormFinder (SV)**	**Bestkeeper (SD)**	**geNorm (MV)**	**Delta-Ct (SD)**	**NormFinder (SV)**	**Bestkeeper (SD)**	**geNorm (MV)**
1	*S5* (0.58)	*S5* (0.321)	*S18* (0.73)	*S18|S5* (0.264)	*AK* (0.68)	*AK* (0.178)	*S18* (0.56)	*EF-1|RPS5* (0.182)
2	*PLA2* (0.62)	*PLA2* (0.368)	*EF-1* (0.84)	*S18|S5* (0.264)	*S5* (0.7)	*EF-1* (0.286)	*S28* (0.80)	*EF-1|RPS5* (0.182)
3	*EF-1* (0.65)	*EF-1* (0.44)	*S5* (0.85)	*S28* (0.328)	*EF-1* (0.71)	*S5* (0.33)	*S5* (0.89)	*AK* (0.242)
4	*S28* (0.67)	*S28* (0.507)	*S28* (0.89)	*EF-1* (0.437)	*S28* (0.76)	*PLA2* (0.362)	*EF-1* (0.95)	*S28* (0.286)
5	*S18* (0.68)	*S18* (0.534)	*PLA2* (1.23)	*PLA2* (0.528)	*PLA2* (0.84)	*S28* (0.533)	*AK* (0.99)	*S18* (0.344)
6	*GAPDH* (0.71)	*GAPDH* (0.549)	*GAPDH* (1.30)	*GAPDH* (0.596)	*S18* (0.92)	*S18* (0.813)	*PLA2* (1.25)	*PLA2* (0.44)
7	*AK* (0.73)	*AK* (0.56)	*AK* (1.46)	*AK* (0.645)	*GAPDH* (1.31)	*GAPDH* (1.129)	*GAPDH* (2.13)	*GAPDH* (0.717)
8	*ACT* (0.78)	*ACT* (0.632)	*ACT* (1.51)	*ACT* (0.678)	*ACT* (1.62)	*ACT* (1.554)	*ACT* (2.44)	*ACT* (0.942)

**Table 3 ijms-23-14371-t003:** The values for gene expression stability derived from Best-analysis Keepers of the eight candidates for reference genes. The coefficient of variation is represented by the notation CV [% Ct] ^a^. The value that is denoted by GM (Ct) ^b^ is the geometric mean of the Ct values. The coefficient of determination is denoted by the notation CD **[r^ 2]**
^c^.

**Rank**	**Line 1**					**Line 2**				
	**Gene**	**CV [% Ct] ^a^**	**GM (Ct) ^b^**	**CD [r^2] ^c^**	** *p* ** **Value**	**Gene**	**CV [% Ct]**	**GM (Ct)**	**CD [r^2]**	** *p* ** **Value**
1	*S18*	1.05	19.01	0.736	0.003	*S18*	0.39	19.15	0.945	0.001
2	*ACT*	1.08	20.62	0.910	0.001	*PLA2*	0.57	26.49	0.891	0.001
3	*S5*	1.11	30.92	0.854	0.001	*AK*	0.68	32.59	0.927	0.001
4	*AK*	1.21	28.86	0.924	0.001	*EF-1*	1.06	33.07	0.927	0.001
5	*PLA2*	1.69	26.77	0.976	0.001	*S5*	1.22	31.28	0.801	0.001
6	*GAPDH*	3.17	19.36	0.980	0.001	*ACT*	2.00	22.83	0.980	0.001
7	*EF-1*	2.16	32.46	0.982	0.001	*GAPDH*	2.34	20.23	0.882	0.001
8	*S28*	5.25	23.55	0.937	0.001	*S28*	2.56	24.66	0.904	0.001
**Rank**	**Line 3**					**Line 4**				
	**Gene**	**CV [% Ct]**	**GM (Ct)**	**CD [r^2]**	** *p* ** **Value**	**Gene**	**CV [% Ct]**	**GM (Ct)**	**CD [r^2]**	** *p* ** **Value**
1	*S28*	0.93	23.61	0.830	0.001	*GAPDH*	1.87	17.63	0.910	0.001
2	*ACT*	1.85	22.15	0.970	0.001	*S5*	1.35	30.93	0.689	0.006
3	*S18*	2.26	18.40	0.982	0.001	*S18*	2.59	19.08	0.891	0.001
4	*PLA2*	2.78	25.42	0.980	0.001	*PLA2*	2.41	23.30	0.824	0.001
5	*S5*	2.26	31.45	0.972	0.001	*ACT*	3.10	19.70	0.933	0.001
6	*AK*	2.48	32.63	0.972	0.001	*S28*	2.93	24.08	0.895	0.001
7	*EF-1*	2.75	30.60	0.964	0.001	*AK*	2.74	27.30	0.953	0.001
8	*GAPDH*	4.30	19.78	0.976	0.001	*EF-1*	3.40	30.78	0.990	0.001
**Rank**	**Line 5**					**Integrated sample**				
	**Gene**	**CV [% Ct]**	**GM (Ct)**	**CD [r^2]**	** *p* ** **Value**	**Gene**	**CV [% Ct]**	**GM(Ct)**	**CD [r^2]**	** *p* ** **Value**
1	*ACT*	1.14	18.56	0.819	0.001	*S28*	3.46	23.41	0.947	0.001
2	*PLA2*	0.90	23.40	0.958	0.001	*S18*	4.79	19.8	0.989	0.001
3	*S28*	1.58	24.17	0.856	0.001	*EF-1*	3.16	31.06	0.987	0.001
4	*S5*	1.62	30.97	0.982	0.001	*S5*	3.25	32.14	0.985	0.001
5	*EF-1*	1.74	30.13	0.895	0.001	*PLA2*	4.84	24.44	0.99	0.001
6	*AK*	1.88	29.63	0.974	0.001	*GAPDH*	9.64	18.84	0.993	0.001
7	*S18*	4.71	19.06	0.982	0.001	*AK*	6.15	31.19	0.982	0.001
8	*GAPDH*	7.62	16.50	0.966	0.001	*ACT*	12.07	22.6	0.987	0.001
**Rank**	**Ovary**					**Fat body**				
	**Gene**	**CV [% Ct]**	**GM (Ct)**	**CD [r^2]**	** *p* ** **Value**	**Gene**	**CV [% Ct]**	**GM (Ct)**	**CD [r^2]**	** *p* ** **Value**
1	*S28*	2.23	23.66	0.50	0.001	*S18*	2.49	18.94	0.137	0.012
2	*S5*	2.35	31.69	0.51	0.001	*S5*	1.65	31.11	0.477	0.001
3	*EF-1*	2.49	31.25	0.35	0.001	*S28*	2.70	24.01	0.228	0.001
4	*PLA2*	3.74	24.73	0.05	0.138	*EF-1*	3.99	31.39	0.689	0.001
5	*S18*	5.19	19.37	0.51	0.001	*GAPDH*	7.28	18.65	0.887	0.001
6	*GAPDH*	6.96	19.40	0.72	0.001	*ACT*	6.70	20.71	0.814	0.001
7	*ACT*	7.04	19.71	0.67	0.001	*PLA2*	5.84	25.03	0.724	0.001
8	*AK*	5.70	29.49	0.53	0.001	*AK*	6.49	30.13	0.601	0.001
**Rank**	**Thorax**					**Head**				
	**Gene**	**CV [% Ct]**	**GM (Ct)**	**CD [r^2]**	** *p* ** **Value**	**Gene**	**CV [% Ct]**	**GM (Ct)**	**CD [r^2]**	** *p* ** **Value**
1	*S18*	3.60	20.31	0.288	0.001	*S18*	2.73	20.62	0.663	0.001
2	*EF-1*	2.77	30.48	0.632	0.001	*S28*	3.39	23.64	0.208	0.002
3	*S5*	2.59	32.74	0.280	0.001	*S5*	2.70	33.09	0.734	0.001
4	*S28*	4.00	22.36	0.426	0.001	*EF-1*	3.04	31.11	0.697	0.001
5	*PLA2*	5.08	24.09	0.602	0.001	*AK*	3.00	32.94	0.733	0.001
6	*GAPDH*	7.67	16.90	0.558	0.001	*PLA2*	5.23	23.89	0.796	0.001
7	*AK*	4.52	32.35	0.464	0.001	*GAPDH*	10.30	20.58	0.837	0.001
8	*ACT*	6.34	23.71	0.501	0.001	*ACT*	9.01	26.97	0.826	0.001

**Table 4 ijms-23-14371-t004:** A comprehensive ranking of the stability values of the eight reference genes. (SV) ^a^ indicates the stability value of the genes. (MS) ^b^ represents the maximum stability value.

**Rank**	**Line 1**			**Line 2**			**Line 3**			**Line 4**			**Line 5**		
	**Comprehensive Ranking**			**Comprehensive Ranking**			**Comprehensive Ranking**			**Comprehensive Ranking**			**Comprehensive Ranking**		
	**Gene**	**(SV) ^a^**	**(MS) ^b^**	**Gene**	**(SV)**	**(MS)**	**Gene**	**(SV)**	**(MS)**	**Gene**	**(SV)**	**(MS)**	**Gene**	**(SV)**	**(MS)**
1	*AK*	2.21	*AK*	*EF-1*	1.86	*EF-1*	*PLA2*	1.41	*PLA2*	*S18*	1.86	*S18*	*S5*	2.00	*S5*
2	*PLA2*	2.34	*PLA2*	*AK*	1.86	*AK*	*S5*	2.11	*S5*	*ACT*	2.11	*ACT*	*ACT*	2.45	*ACT*
3	*S5*	2.78		*PLA2*	2.21		*ACT*	3.6		*PLA2*	2.83		*PLA2*	2.66	
4	*GAPDH*	3.22		*S18*	4.30		*S18*	3.83		*GAPDH*	3.22		*AK*	3.46	
5	*ACT*	3.94		*ACT*	4.36		*S28*	4.76		*S28*	4.24		*S28*	3.46	
6	*S18*	3.96		*S5*	5.23		*EF-1*	5.45		*S5*	4.7		*EF-1*	3.87	
7	*EF-1*	5.86		*GAPDH*	5.63		*GAPDH*	5.62		*AK*	5.92		*S18*	7.00	
8	*S28*	8.00		*S28*	8.00		*AK*	5.63		*EF-1*	8.00		*GAPDH*	8.00	
**Rank**	**Ovary**			**Fat body**			**Thorax**			**Head**			**Integrated sample**		
	**Gene**	**(SV)**	**(MS)**	**Gene**	**(SV)**	**(MS)**	**Gene**	**(SV)**	**(MS)**	**Gene**	**(SV)**	**(MS)**	**Gene**	**(SV)**	**(MS)**
1	*S18*	1.97	*S18*	*EF-1*	1.41	*EF-1*	*S5*	1.32	*S5*	*AK*	1.97	*AK*	*S18*	1.41	*S18*
2	*EF-1*	2.28	*EF-1*	*ACT*	2.21	*ACT*	*S18*	2.24	*S18*	*S5*	2.06	*S5*	*S5*	1.68	*S5*
3	*S5*	2.38		*PLA2*	3.71		*EF-1*	2.91		*EF-1*	2.21		*S28*	2.63	
4	*PLA2*	2.83		*GAPDH*	4.23		*PLA2*	3.16		*S28*	3.56		*EF-1*	3.22	
5	*S28*	4.30		*S18*	4.30		*S28*	3.72		*S18*	3.66		*PLA2*	5.00	
6	*GAPDH*	5.23		*S28*	4.40		*GAPDH*	6.00		*PLA2*	5.18		*GAPDH*	6.00	
7	*ACT*	6.24		*S5*	4.56		*AK*	7.00		*GAPDH*	7.00		*AK*	7.00	
8	*AK*	8.00		*AK*	8.00		*ACT*	8.00		*ACT*	8.00		*ACT*	8.00	

## Data Availability

Data are contained within the article or Appendix A.

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
