# Peer review of "Evaluation of Reference Genes for Real-Time Quantitative PCR Analysis in Tissues from Bumble Bees (*Bombus Terrestris*) of Different Lines"

_ijms, 2022, doi:10.3390/ijms232214371_

Round 1
Reviewer 1 Report
Sankar and co-workers investigated “Evaluation of reference genes for real-time quantitative PCR analysis in tissues from bumble bee (Bombus terrestris) of different lines”. This study reveals that gene expression studies have become extremely important in understanding gene function and molecular mechanisms. In this state (Bombus terrestris) species to test the expression stability of eight reference genes (ACT, AK, PLA2, EF-1, S5, S18, S28 and GAPDH) in RT-qPCR procedures in five different lines and tissues. The topic is interesting. However, the manuscript requires a minor revision to be accepted for publication Concerning the writing aspects, first of all, to go into the details, the authors should change the genes name bond style in italic. It is incomplete in some places. It would be concise and more clearly stated. Though the study is interesting, there are minor issues that need to modify, details are as follows:
.
Title:
The title is clearly suitable for this manuscript but needs to change the bond style as follows
Line 2 to 3- Species name bond style should be changed to italic “Evaluation of reference genes for real-time quantitative PCR analysis in tissues from bumble bee (Bombus terrestris) (Bombus terrestris) of different lines”
ABSTRACT:
The author clearly expressed and presented in this part of the abstract overall very good
Introduction:
The author clearly expresses and presents in this part of the Introduction
Materials and Methods:
Line 91- The author uses one target gene (Vg) for validation, but the author mentioned “two” need to change the word (two) “one”
Line- 113- The author should check diluted DNA Concentration 250ng/ul or 100
Line -124- The author mentioned Table 1; TM 59○C but materials and methods parts present different TM I think 58○Cchange to (59 ○C)
Line 156- Author should be changed R2 (R2)
Results:
Line 186- Author should change “coefficients (R2 > 0.99) (R2 > 0.99)”
Line 213- gene name S5 should be italic S5 (S5)
245- gene name should be italic AK, S18, EF-1, S28 (AK, S18, EF-1, S28)
Line 260, Line 264- PLA2, AK should be changed bond italic (AK, PLA2)
Line 280, Line 281- AK S5 should be changed bond italic (AK, S5)
Line 299 to 300- genes (P = 0.997 for lines 4 and 0.952 for line 5) genes “(for lines 4 and “P= 0.952” for line 5)”
Discussion:
Line 314-Author should be changed this sentence “the honey bee across different lines” (the “bumble bee” across different lines)
Line 343- The author mentioned the four algorithms, but this line presents only three algorithms “NormFinder, BestKeeper, and geNorm” Author need to add “(Reffinder, NormFinder, BestKeeper, and geNorm)”
Line 187- The author should change the suitable sentence (gene expression of target genes throughout the reprogramming “experimental” process).
Funding:
Line 394- Missing this word; This “work” was supported
References:
The author should remove these references before finalizing now this reference is correct position, but no need for more because these are already present “Reference no. “(15, 20, 21, 24)”
Line 432 to 433 - “15. Vandesompele, Jo, et al. "Accurate normalization of real-time quantitative RT-PCR data by geometric averaging of multiple internal control genes." Genome biology 3.7 (2002): 1-12.”
Line 442 to 443 - “20. Bustin, Stephen A., et al. "The MIQE Guidelines: Minimum Information for Publication of Quantitative RealTime PCR Experiments." (2009): 611-622.”
Line 444 to 445- “21. Cassan-Wang, Hua, et al. "Reference genes for high-throughput quantitative reverse transcription–PCR analysis of gene expression in organs and tissues of Eucalyptus grown in various environmental conditions." Plant and Cell Physiology 53.12 (2012): 2101-2116.”
Line 452 to 453- “24. Scharlaken, Bieke, et al. "Reference gene selection for insect expression studies using quantitative real-time PCR: The head of the honeybee, Apismellifera, after a bacterial challenge." Journal of insect Science 8.1 453 (2008): 33”
Table:
Table1. The author should change all genes bond italic ACT, AK, PLA2, EF-1, S5, S18, S28, GAPDH “(ACT, AK, PLA2, EF-1, S5, S18, S28, GAPDH)”. Please check all.
Author Response
Dear sir greetings of the day!! thank you very much for taking the time to review my materials, For your valuable and insightful comments, which significantly helped to improve the manuscript’s quality. The reviewer’s suggestions were taken seriously and the necessary discussions, corrections, and modifications have been included in the revised manuscript
Here I have attached a file for your comment and consideration..... thank you once again
Thank you
Sincerely
Kathannan sankar

Reviewer 2 Report
The manuscript entitled "Evaluation of reference genes for real-time quantitative PCR 2 analysis in tissues from bumble bees (Bombus terrestris) of different lines” by Sankar et al. evaluated several housekeeping genes in different bumble bee lines and different tissues using target gene Vg and four programmers (RefFinder, NormFinder, BestKeeper, and geNorm), and results showed that S28, 21 S5, and S18 ribosomal protein genes and the PLA2, EF-1 genes had the highest stability. The study provided more information on the qPCR normalization strategy to make the relative qPCR results more reliable in bumble bees. Authors collected samples from 5 bumblebee virgin queens and heads, thorax, ovary and fat body within 24 h post emergence. However, the validity of a reference gene is highly dependent on the experimental conditions, so I think authors should set up more conditions such as different development stages, different time points post infection treatment, or other treatments to validate those reference genes. And there are some comments listed below:
1. Why the target gene is Vg? Vg is mainly expressed in the fat body in insects for egg formation, and highly expressed after mating, which means Vg is related to reproduction, you should set up more time points if you want to validate the reference genes during reproduction. Please choose more target genes from different physiological pathways to test these references, then readers will know which ref fits their experiments. And I think that’s the most important goal of the manuscript.
2. The resolution of figures 7, 8, 1, and 3, 4, 6 is low, please replace them with a higher one.
3. Figure order is wrong, and some figures are in Methods, please correct them and add more information in the figure legend.
4. Line109 is hard to follow, please figure it out.
5. When you use different programs to test these reference genes, I found the resulting stabilities are inconsistent, so how could you conclude that EF-1, S5, S18, and GAPDH would be the most 357 suitable as optimal reference genes for the normalization of the target gene expression? These programs just give you suggestions, but the reality may be totally different, so please set up qPCR to double-check using different target genes and different treatments at different time points.
Author Response
We thank the referees for their valuable and significant comments that helped us to improve the manuscript.

Round 2
Reviewer 2 Report
Thanks for your update, I agree with the modifications of the manuscript.